# Towards the Contemporary Conservation of Cultural Heritages: An Overview of Their Conservation History

**Lanxin Li** * **and Yue Tang** 

Department of Architecture and Built Environment, Faculty of Engineering, University of Nottingham, Nottingham NG7 2RD, UK; yue.tang@nottingham.ac.uk
* Correspondence: lanxin.li@nottingham.ac.uk

**Abstract:** This paper seeks contemporary cultural heritage conservation principles by reviewing its history, starting from the 18th century, in practices, international documents, and the literature on this topic. It intends to lay a foundation to avoid damaging cultural heritages by misconducting conservation. This study first found that the conservation objects of cultural heritage include particularly important ones and general ones that are closely related to daily life, and they involve tangible and intangible aspects. Second, cultural heritage conservation involves document-based restoration when necessary as well as identifying the value of them to play their role nowadays. Third, integrating cultural heritage conservation within the context of sustainable development goals is essential for successfully balancing the relationship between the built and natural environments. Fourth, mobilising the public to participate in cultural heritage conservation enables the implementation of conservation to meet the expectations of the public, and may have a positive influence on people's consciousness. Fifth, as a treasure, cultural heritage conservation is a global responsibility that involves conjoint cooperation. Sixth, more cultural heritage conservation interdisciplinary methods have been developed and could be applied, but this should be limited in order to not destruct their authenticity and integrity.

**Keywords:** cultural heritage; conservation history; conservation principles



## 1. Introduction

Cultural heritage comprises the tangible and intangible heritage assets of a group or society that are inherited from past generations [1]. Increasing natural erosion and disasters, modernisation, and social conflicts are intensively threatening cultural heritages [2]. If these assets are destroyed, there will be a loss of human treasures, because, as the products of human social life, they are the record of the development of human civilisation [3] and are significant for contemporary social, economic, and ecological reasons [4].

In terms of their social importance, first, cultural heritages are beneficial to people's physical and mental health, as their unique history often brings typical aesthetic characteristics that are rarely found in contemporary life. People can be attracted by them, enjoy them [5], and even undergo a "peak experience" that may be "ecstatic" [6–8]. Meanwhile, cultural heritages may be associated with certain people's memories, as they often provide the context for people's attachment to a place [9]. Hence, people who access objects that carry their personal memories tend to have a sense of belonging and peace of mind. These "ecstatic" and "belongingness and peacefulness" are beneficial to people's health [6–8]. Second, cultural heritages help regions to maintain social and territorial cohesion. The inheritance of the culture carried within them enables people who share a common culture to identify with each other, increasing regional social solidarity [10]. They can potentially promote access to and enjoyment of cultural diversity, enabling different cultural groups to understand each other and leading to a reduction in social conflicts caused by cultural shock [11]. Third, cultural heritages are an important vehicle for the

transmission of experiences, skills, and knowledge between generations; they are also a source of inspiration for creativity and innovation that generate contemporary and future civilisation [11]. In sum, cultural heritages can increase people's health, maintain social cohesion, and transfer knowledge.

In terms of their economic importance, cultural heritages can diversify the economy, supporting economic growth. The cultural variety carried by them can inspire innovation for improving productivity. And, their diversity means that their re-utilisation range is extensive, which allows the cultivation of businesses of different types [11,12].

In terms of their ecological importance, many cultural heritages carry ancestral ecological wisdom about how humans and nature can coexist harmoniously [13]. Such wisdom can not only be referenced for current urban development but also conserved for coordinating contemporary relationships between humans and nature [13].

Therefore, cultural heritages contribute to contemporary life, reasoning the necessity for sustainable conservation. Their conservation has gradually received more attention since the 18th century. Following the industrial revolutions and the two World Wars, with significant damages to cultural heritages throughout the world, conservation practices gradually increased. In the mid-20th century, many international organisations like the United Nations Educational, Scientific and Cultural Organisation and the International Council on Monuments and Sites issued numerous international documents about cultural heritage conservation. These documents have been gradually refined, along with changes in global situations. They witnessed the development of international cultural heritage conservation and provided theoretical guidance. Simultaneously, a considerable literature about cultural heritage conservation in more specific areas is gradually blooming and extending to conservation research topics.

However, insufficient knowledge arising from unfamiliarity with the conservation experience leads to a number of issues. First, the objects needing conserving are not clearly defined, resulting in many cultural heritages being neglected and even destroyed.

Second, the relationship between conservation and utilisation has not been well balanced. As a result, some cultural heritages could not be utilised whilst carrying on their role in contemporary life. They have missed the opportunity to become an economic source to support their continuous conservation due to a lack of vitalisation and difficulty being accessed [14,15]. In contrast, certain places substitute tourism development for conservation, a practice in which cultural heritages are over-utilised and destroyed by unlimited exploitation and uncontrollable destruction [15].

Third, the scope of cultural heritages that need to be conserved is unclear, increasing the risks of destruction to the surrounding environment, which nurses and protects them [15]. Any changes in the surrounding environment would significantly impact cultural heritages and their conservation area. Furthermore, in certain areas, conservation is considered to be contradictory to modern urban development, and, therefore, developments surrounding cultural heritages are gradually impacting them. Instead, conservation should be included in the macro-planning perspective in order to pursue a win–win situation between conservation and urban development [16].

Fourth, in many regions, the conservation of cultural heritages is often dominated by experts and lacks public participation, triggering many problems, as follows [14]: (1) Some experts lack sufficient scientific conservation knowledge, meaning that the attempt to conserve cultural heritages may result in their destruction. For example, some experts over-restore cultural heritages and may destroy some of the original historical information carried by them [17]. Another example of this is that, to ensure a unity and integrity of style, a real, although damaged, heritage artifact is replaced by a new but fake copy [14,15], which also destroys historical information. (2) Expert-oriented conservation cannot mobilise the whole of society behind said conservation efforts. This kind of conservation often does not fully consider the values and interests of other stakeholders, ignoring their rights, especially those of the general public, whose members have relatively less voice [14]. If the wishes of other stakeholders are ignored, public enthusiasm for conservation decreases, making it

hard to give the public a role in conservation practices in areas such as daily maintenance, where the contribution of experts is limited. (3) Without the involvement of indigenous people in the conservation of cultural heritages closely related to their daily life, the values held by these groups may be ignored by conservation practices. This may lead indigenous people to move out, draining or extinguishing the intangible cultural heritages carried by them [18]. (4) A lack of public participation in cultural heritage conservation may decrease public cultural understanding and awareness of the importance of conservation efforts, meaning that, when conservation clashes with people's other interests, conservation tends to be relinquished [14,15,19]. (5) The public cannot participate in conservation practices on terms that are fair and just, which may increase social injustice and trigger many social conflicts [20].

Fifth, the fact that the diversity of cultural heritages means that some of them may not be conserved fairly, resulting in further problems, as follows: (1) Some cultural heritages may be conserved following the conservation method of a dominant cultural heritage, which may lack sufficient consideration of specific contexts and characteristics and, therefore, be unsuitable. (2) There may be a neglect of certain cultural heritages of weak groups who tend to lack the economic and technological power to conserve them, resulting not only in the loss of world treasure but also in social problems. As a result, people whose memory is associated with these heritages may lose their emotional support, which may damage their self-esteem and mental health [21]. Meanwhile, such people may also think that their culture cannot be accepted equally and inclusively, which may lead to discrimination, affecting social stability [22].

Sixth, possible methods of conservation and how to use them are unclear. Hence, some available scientific technologies and methods may be ignored, triggering the destruction of historical information [23].

In order to solve the problems encountered in the conservation of cultural heritages, it is necessary to learn from history and take in experiences from the past. Although some research has paid attention to the conservation history of cultural heritages, the reviewed history is fragmented and lacks close logical connections based on a continuous timeline [14,24,25]. Moreover, the reviewed history mainly concentrates on pre-2010 periods, lacking attention to recent years and a deeper thinking of the inspiration that can be drawn from history [14,24,25]. Contemporary society is witnessing a rapid development in science and technology and increasingly fierce conflicts between human beings and ecology, as demonstrated by the COVID-19 pandemic and the frequent occurrence of extreme climate events [26]. These situations have profoundly affected the development of conservation practices for cultural heritages. Therefore, in order to make up for the insufficiency of the existing research and provide basic principles for the contemporary and future conservation of cultural heritages, this paper reviews cultural heritage conservation history based on practices, international documents, and the literature on the topic.

## 2. The Development of Cultural Heritage Conservation

### 2.1. Conservation in the Early 17th–18th Centuries

Cultural heritage conservation in the modern sense began with the Enlightenment [27]. During this period, thinkers advocated rationality, believed in the domination of science over nature, and advocated liberating people from the irrationality of myth, religion, and superstition. They believed that universal, eternal, and unchanging truth could be revealed only in this way [24]. The ideas of the Enlightenment had a huge influence on the European continent. Hence, during that time, young men were exposed to Greek and Roman history across Europe because ancient Greek and Roman culture advocated rationality [24]. Consequently, the Grand Tour became a highly desirable way for aristocrats and gentry to polish their sons' education by visiting Rome, Naples, Venice, Florence, etc. The tour inspired many travellers to take a greater interest in Roman history and art [28], and therefore, in archaeology, with its associated historical information and cultural heritage. Hence, archaeological work achieved remarkable results [24], and the unearthed Greek and

Roman cultural heritages were transported to museums. At the same time, people also had unprecedented enthusiasm for the cultural heritage of architecture and paid close attention to conserving and restoring it. In general, the conservation at that time was mainly led by the elite who could access education and participated in the field of archaeology [24].

*2.2. 19th-Century Conservation Debates*

2.2.1. The Debate between Restoration and Anti-Restoration

Since enthusiasm for cultural heritage conservation in the 18th century was mainly aroused by visual experiences, pursuing the visual style of cultural heritages was the main topic at the time. This set off a wave of "stylistic restoration" of conserving architectural heritages. The French architect Eugene Viollet-Le-Duc (1814–1879) is a representative person [25]. He stated that "to restore a building is not to preserve it, to repair, or rebuild it; it is to reinstate it in a condition of completeness based on imaging original architect' thought which could never have existed at any given time" [29,30]. Based on this principle, many architectural restorations in this time were trying to search for traces left by the original architect and strived to completely restore architecture to the form of their previous era [24]. A representative example is the restoration of Notre Dame Cathedral, mainly promoted by Viollet-Le-Duc in the mid-19th century, in which the originally destroyed elements of the architecture, like statues and glass, were replaced by newly designed elements [31]. Although he restored the integrity and shaped the artistry of the cathedral, this approach had subjective creation, which destroyed the authenticity of historical materials to a certain extent [24,32].

Influenced by the tendency of writers of British Romantic literature (1830s–1890s) to be dissatisfied with the development of capital-oriented urban civilisation and eager to retreat to nature [33], British art critic John Ruskin (1819–1900) and others praised the natural beauty of ruins and strongly opposed the practice of "stylistic restoration", arguing that restoration was a destruction of a building, resulting in the loss of historical authenticity [25]. They emphasised the charm and value brought by the age of buildings, and believed that no matter how dilapidated historic monuments were, they should be kept as they were and left to future generations [21]. They advocated replacing restoration with protection to preserve all historical information on cultural heritages [24]. Although this approach indeed protects historical information, it has resulted in the demise of a large number of cultural heritages [24,32].

This dispute between restoration and anti-restoration was gradually resolved with the development of Italian conservation thought, which advocated restoration based on history. Camillo Boito (1836–1914) was one of the founding figures [34], and believed that architectural heritages should be regarded as historic documents, each part of which reflects history. He advocated that the status quo of cultural relics must be respected, and their restoration needs to be based on historical evidence. Since it is necessary to reinforce rather than add anything, restoration must not change the appearance of a building from the era when it was created. Also, all changes that have been made must be recorded and conserved [24,25,32]. This kind of restoration is also a continuation of the artistry of cultural heritages [35]. This view continued into the 20th century and influenced the formulation of the "*Athens Charter for the restoration of historic monuments*" (1931) [36].

The "*Athens Charter for the restoration of historic monuments*" was the first international document to encourage modern conservation, and propose restoration principles. It recommended that when, as the result of decay or destruction, restoration appears to be indispensable, the historic and artistic work of the past should be respected without excluding the style of any given period [25]. The judicious use of modern materials for the consolidation of ancient monuments, a kind of cultural heritage, is allowed. This consolidation should, whenever possible, be concealed to preserve the character of the restored monument [36]. Later, the "*Venice Charter*" (1964) [37] further pointed out that all the restoration contributions of each era need to be respected. New technologies can

be reasonably used if necessary; any unavoidable additions must be distinguished from heritages. This restoration principle had a profound impact on the future.

2.2.2. The Contradiction between Urban Modernisation and Cultural Heritage Conservation

The Enlightenment provided the ideological foundation for two Industrial Revolutions. To make urban space suitable for large-scale industrial production, urbanisation was promoted, threatening cultural heritage conservation [24]. However, conservation was not yet aligned with the overall goals and trajectory of urban development in the 19th century [24], so many cultural heritages were swallowed by urbanisation.

Haussmann's renovation of Paris is a classic example. Industrial development brought people together in Paris in the 19th century, and the huge population load made Paris increasingly uninhabitable. In terms of hygiene, there was less and less open space because of the disorderly construction of private buildings. Living spaces lacked air, light, and space for walking activities, which affected people's health. Meanwhile, without enough space, garbage accumulated in living spaces, causing an epidemic of infectious diseases [38]. In terms of transportation, most of the narrow streets built in the Middle Ages were only suitable for walking, which restricted urban traffic [38]. In terms of security, the intricate and narrow streets provided spaces that were hard to manage, breeding social disturbances and crimes. How to make a city sustainable for modern life became a thorny issue [38]. Hence, George-Eugène Haussmann initiated the renovation of Paris from 1853 to address these problems [39].

The renovation mainly involved the construction of roads, houses, and infrastructure. The total length of various roads in Paris was 239 miles in 1852, increased to 261 miles in 1860, and to 525 miles when Haussmann finished his work. The roads were also much wider than before, more than doubling from about 39 feet to 79 feet [38]. The house renovation aimed to solve the housing problem. A large number of old houses were replaced by new ones, which could cater to more people's living needs. From the end of 1852 to the end of 1859, 4349 houses in Paris were demolished, and 9617 new houses were supplied to the market, a net increase of 5268 [38]. Infrastructure in addition to roads was set up, including lampposts, public toilets, benches, awnings, pavilions, garbage cans, and fountains, and public buildings such as city halls, schools, churches, and hospitals in the city were improved through reconstruction, repair, and new construction [38].

The renovation greatly improved the quality of life for people in Paris, and a series of health, traffic, and safety problems were greatly improved [38]. However, from the perspective of cultural heritage conservation, this type of renovation changed the organic structure of the traditional city. On the one hand, the change ensured the possibility that the traditional city could continuously develop and maintain vitality in the new situation. On the other hand, it destroyed many traditional urban spaces and the humanistic relationships contained in the spaces [24]. The event promoted the way for considering cultural heritage conservation from an urban perspective and influenced the formulation of the "*Athens Charter*" (1933) [40], which proposed properly conserving ancient buildings that do not harm the health of residents.

*2.3. 20th-Century Contributions to Conservation*

2.3.1. Cultural Heritage Conservation from Urban Functionalism to Urban Regeneration

Under the influence of the idea of functional rationality guided by the Industrial Revolutions [41], the modern movement begun by the Bauhaus in 1919 [42,43], advocated that form should follow function (functionalism), embrace minimalism, and reject ornament. Regarding urban development, a series of problems were increasingly emerging, such as population growth, traffic congestion brought by new vehicles, and lack of recreational space. In order to solve these problems, urban planning also began to be function-orientated. The "*Athens Charter*" [40] advocated rational zoning of urban functions into separate residential, industrial, or commercial areas to support population growth, ease traffic, and improve living quality. The charter also based the conservation of historic buildings on

zones. It advocated that under all possible conditions, arterial roads should avoid passing through areas with ancient buildings [40]. This isolated conservation of cultural heritages did not integrate them organically with urban development.

However, urban functionalism gradually brought inconvenience to people's lives because various functions were far apart and made the city devoid of vitality. In the 1960s, opponents of rational planning advocated that living space should be pleasant. The "*Charter of Machu Picchu*" (1977) [44] formally criticised and supplemented the "*Athens Charter*" [40] to advocate the pleasantness of living space, which was mainly reflected in diversifying the functions of every area to facilitate the convenience of residents' life, and maintaining the cultural context of places to enrich people's spiritual world.

The adaptive reuse of cultural heritages can achieve the idea of the pleasantness of living space to a certain extent. Jacobs [12] called on people to pay attention not only to critical historic buildings but also to ordinary old buildings when conserving cultural heritages. First, the prices of old buildings are relatively low, attracting development from businesses that can not afford high prices. Thereby a variety of functions can be cultivated in the same area, which can meet the needs of residents, improve the convenience of people's lives, and bring vitality to the city. Secondly, the unique characteristics of old buildings can give people a pleasing view, preserve people's memories, and endow the urban context with a historical perspective [12]. Then, in the "*Venice Charter*" [37] the conservation objects of cultural heritages were officially expanded from important monuments to traditional settings. This view allowed reasonable reuse of cultural heritages to be made for some socially useful purpose, limited only by the need to preserve their authenticity [45]. This gives cultural heritages a function in the life of the community, enabling their conservation to be integrated with comprehensive urban regeneration, thus benefiting regeneration-oriented urban development and preserving cultural heritages of various eras for future generations.

With the connection between cultural heritage conservation and urban development gradually becoming closer, the conservation objects began to expand to historic cities, towns, and villages from the 1970s, as shown in the "*Washington Charter*" (1987) [46], the "*Bruges Resolution*" (1975) [47], and the "*Nairobi Recommendation*" (1976) [48]. The phenomenon can mainly be attributed to the following. On the one hand, with further urbanisation, a large number of people moved into the city, resulting in the abandonment or decline of some historic cities, towns, and villages. In contrast, a large amount of new economic activity in other historic cities, towns, and villages tended to destroy the original environment.

Historic cities, towns, and villages are closely related to residents' daily life, so their conservation is closely associated with public needs. If these needs are to be met and residents are enabled to continue living there to conserve the intangible cultural heritages they carry, it has gradually become clear that public participation in the conservation is important. Without public participation, gentrification may occur. An example of this occurred in the Marais district of Paris, France in the 1960s [49]. Because one of the royal residences was located there from the 14th century, many private townhouses were built in the area during the 16th–18th century [49]. When the king's residence shifted to the Louvre and then to Versailles, the aristocracy left the area and its buildings were progressively neglected [49]. Then, with the dramatic population increase promoted by industrialisation, the area was gradually occupied by many residents and became "one of the most densely populated slums" [49]. The slum was gradually turned into "one of the most fashionable districts" from the 1970s because the old buildings of the area were adaptively reused by developers for boutique retail [49]. Although the reuse found a contemporary function for the ancient buildings and promoted their sustainable conservation, the ordinary residents in the conserved heritages were replaced by the bourgeoisie, because residents could not afford the living expenses there. Gentrification often easily leads to the problem that the original residents have nowhere to live, and also results in the loss of the culture carried by the original residents.

During the same period, in contrast, the conservation of cultural heritage in Bologna, Italy, gave residents' living standards the same importance as heritage conservation, and promoted the participation of the people in the decision-making process [50]. It reversed the situation that the conservation of cultural heritages was the privilege of the elite, and created a pleasant living urban environment for the public while mobilising the forces of the whole society behind the conservation to express their needs [50]. It also enabled the residents to continue to live there, and contributed to conserving the culture and folklore carried by the residents [51], which is an important part of the urban context. Public participation in cultural heritage conservation has been emphasised by many international documents such as the "*Recommendation concerning the protection, at national Level, of the cultural and natural heritage*" (1972) [52], the "*Nairobi Recommendation*" [48], and the "*Charter for the protection and management of the archaeological heritage*" (1990) [53]. Meanwhile, in order to mobilise the public's enthusiasm for participating in cultural heritage conservation and enhance the scientific nature of public conservation, educating the public has also begun to be emphasised, as shown in the "*Guidelines for education and training in the conservation of monuments, ensembles and sites*" (1993) [54]. These documents demonstrate that urban regeneration has begun to move towards public participation in cultural heritage conservation.

Additionally, beginning in the 1960s, the objects of cultural heritage conservation were also expanded to industrial heritages [24]. Industries began to encounter resource and ecological limits at this time, so many of them declined and were abandoned. Meanwhile, energy crises and economic downturns caused people to pay more attention to the adaptive reuse of abandoned industries to save expenses [24]. For example, the abandoned factory area south of Houston Street in New York, USA, was transformed into the SOHO business district by artists in the 1960s [55]. After a decade, some abandoned old wharves in the USA, such as Baltimore Wharf and Fishermans Wharf, were reused as spaces with commercial and recreational functions [24]. In the 1990s, after the Ruhr industrial area in Germany was abandoned, it was transformed into an industrial tourist attraction [24]. Subsequently, international documents on conserving industrial heritages have also been promulgated, including the "*Nizhny Tagil Charter*" (2003) [56] and the "*TICCIH Principles*" (2011) [57].

In general, cultural heritage conservation has evolved from only restoring cultural heritages to exerting their value in contemporary society. Conservation objects have expanded from ones traditional in the field of archaeology to more general ones related to daily life. Also, conservation has gradually developed from elite conservation to mass conservation based on the public's daily needs. All of these developments demonstrate that conserving cultural heritages has been integrated with urban regeneration. These changes make it important to identify and evaluate what objects are cultural heritages that need to be conserved and what their values are in society, according to the "*Barra Charter*" (1999) [58,59]. Based on such evaluation, conservation strategies can be scientifically formulated to better play their part in the city of today and the future.

2.3.2. Cultural Heritage Conservation in the Context of Globalisation

In the two world wars during the first half of the 20th century, considerable cultural heritages suffered looting and destruction. The phenomenon drew the attention of the international community, and people gradually realised that cultural heritages are the precious wealth of all humankind. Subsequently, UNESCO adopted the "*Convention for the protection of cultural property in the event of armed conflict*" (1954) [60], regulating that the parties to the agreement shall respect and protect the cultural heritages of each country and shall not deliberately attack the cultural heritages of other countries. The protection of cultural heritages has transcended the borders of countries [61].

In 1959, Egypt (then the United Arab Republic) turned to UNESCO for large-scale financial, scientific, and technical assistance to protect the historic treasures of the Nubian ruins, which were threatened by inundation as the construction of the Aswan Dam caused the Nile River to rise. Following the appeal from UNESCO in 1960 to governments, organi-

sations, public and private foundations, and individuals to provide services, equipment, and finance to protect the ruins, an unprecedented international cooperation to save them officially began. More than 50 countries contributed finance and technology. Several important Nubian structures were relocated above the waterline for preservation [62]. The conservation practice prompted the formation of the "*Convention concerning the conservation of the world cultural and natural heritage*" (1972) [63]. This document officially clarified the concept of cultural heritages, including monuments, groups of buildings, and sites. It established the "*World Heritage List*" and encouraged international organisations to help conserve world cultural heritages through international cooperation in terms of financial, artistic, and technical assistance. Then, the "*Recommendation concerning the protection, at national Level, of the cultural and natural heritage*" [52] and the "*Operational Guidelines for the implementation of the World Heritage convention*" (1977) [64] were promulgated to promote international cooperation to conserve the diversity of world cultural heritages.

With the development in conservation of the diversity of world cultural heritages, people gradually realised that the previous conservation was mainly targeted at masonry heritages, widely distributed in the West, but neglected Oriental cultural heritages, mainly composed of wood and wall paintings. Conservation of eastern cultural heritages cannot be directly copied from the methods of the West. The woods and paintings of eastern cultural heritages are easily damaged, and they are inevitably restored by replacing wooden components and reconstructing wall paintings to maintain historical information. Therefore, the stipulation in the international documents represented by the "*Convention concerning the conservation of the world cultural and natural heritage*" [63] that reconstructed buildings cannot be rated as cultural heritages because their lost authenticity is not suitable for the East [65]. Consequently, the "*Nara document on authenticity*" (1994) [66] proposed that the authenticity and value of heritage properties must be considered and judged within their cultural contexts. Meanwhile, a series of international documents concerning the conservation of eastern cultural heritages were promulgated, including the "*Principles for the conservation of wooden built heritage*" (1999) [67], the "*Hoi An Protocols for best conservation practice in Asia: professional guidelines for assuring and preserving the authenticity of heritage sites in the context of the cultures of Asia*" (2005) [68], the "*Beijing Document on the conservation and restoration of historic buildings in East Asia*" (2007) [69], and the "*Principles for the preservation and conservation/restoration of wall paintings*" (2003) [70]. Additionally, establishing a buffer zone around cultural heritages was gradually emphasised, as shown in the "*Xi'an Declaration*" (2005) [71]. The zone can prevent cultural heritages from being destroyed by restricting their surrounding modernisation and conserving their surrounding cultural context and physical environment. When conserving the Eastern heritages, it is particularly important to establish a buffer zone to protect them because they are easily destroyed.

Subsequently, cultural heritages in many regions throughout the world have begun to be paid more specific attention, including Europe [72], North America [20], South America [22], Asia, and Africa [73]. Meanwhile, with the further development of globalisation, people gradually noted that some countries or regions might share a common history, resulting in the cultural heritages in these regions constituting cultural groups or routes, reflecting human historical communication. Hence, the "*ICOMOS Charter on cultural routes*" (2008) [74] was promulgated to encourage people of different regions to cooperatively conserve the authenticity and integrity of cultural heritages over a larger scope. In general, conserving cultural heritages from the global perspective has gradually become mainstream [2].

### 2.4. Conservation around the 21st Century

2.4.1. From Conserving Culture to Playing the Role of Culture

Since the beginning of the 19th century, people have paid attention to conserving the cultural context of cultural heritages. With people gradually understanding cultural heritages, conserving intangible cultural heritages also came to be emphasised because, as a practice, representation, expression, knowledge, or skill, they can reflect historical

information. Intangible cultural heritages consist of non-physical intellectual wealth, such as folklore, customs, beliefs, traditions, knowledge, and language, which are crystals of wisdom created by humans' production and life. Therefore, the "*Convention concerning the conservation of Intangible cultural heritage*" (2003) [75] was promulgated.

With the development of conservation of the culture carried by cultural heritages, people have begun to be aware that culture can influence and even shape people's consciousness, which further affects people's behaviour. First, people accessing culture carried by cultural heritages can increase their understanding of the culture and be attracted by it, which can increase their civic pride. Civic pride, in turn, can increase people's awareness of the importance of conserving the cultural heritages, encouraging them to participate in the conservation, as shown in the "*ICOMOS Charter for the interpretation and presentation of cultural heritage sites*" (2008) [76]. For example, China has established many Archaeological Site Parks in important archaeological sites to provide leisure spaces and to exhibit cultural heritages to people. While people relax there, they can learn historical information and be attracted by history, which arouses public conscious awareness of the importance of conserving them [77].

Second, the culture can promote social stability, reflected in social cohesion, justice, and morality. Regarding maintaining social cohesion, cultural heritages are an important expression of the spirit of the place, which refers to the unique, distinctive, and cherished aspects of a place. It is made up of the spirit of tangible (sites, buildings, landscapes, routes, objects) and intangible elements (memories, festivals, rituals, knowledge, odours), as shown in the "*Québec declaration on the preservation of the spirit of place*" (2008) [78]. Furthermore, the spirit of place constitutes the identity of the place, which refers not only to the distinctiveness of individual places but also to the sameness between different places [79]. The identity of the place enables residents to have a sense of identity with each other, which can increase people's sense of belongingness and decrease social conflicts. These further promote social cohesion [80]. Thus, the spirit of the place brought by cultural heritages also needs to be conserved to strengthen social cohesion. Strengthening social cohesion through cultural heritages has been highly emphasised in recent years. For instance, China conserves the cultural heritages of different ethnic groups to preserve the evidence and memory of their historical communication, which enables people of different ethnic groups to have a cultural identity, and cultivate a sense and spirit of the community of the Chinese nation, so as to enhance the cohesion of the Chinese [81].

In terms of social justice, respecting and fairly conserving cultural heritages created by different people is needed. Cultural heritages can reflect the values, beliefs, and customs of different people [82]. Therefore, conserving cultural heritages is conserving the intangible culture of these people, allowing them to find their identity and increase their cultural confidence. Conserving cultural heritages also enables people associated with these heritages to have a sense of being respected and not being discriminated against, further decreasing social conflicts. The function of cultural heritages in social justice is also emphasised now. For example, in the USA considerable research has been conducted on exploring and conserving cultural heritages related to the history of African immigrants. This conservation increases the sense of cultural belonging and being respected among African descendant populations in North America. Based on their culture, their acculturation can be increased, helping them adapt to their new situation, which can reduce social contradiction and promote regional and global harmony [20,82].

At the same time, cultural heritages associated with good 'morality' have been conserved in order to enable people to understand and honour this morality and use it as their value and norm to regulate their behaviour. For instance, the China government vigorously promotes cultural heritages related to heroes for that people who access these heritages may learn the noble qualities of these heroes, such as hard work [83].

Now, more and more regions are beginning to pay attention to the influence on people's consciousness of culture carried by cultural heritages.

### 2.4.2. Cultural Heritage Conservation in the Context of Sustainable Development

After the 1970s, a series of energy crises broke out, revealing a number of serious problems [84,85]. First, from an economic perspective, energy shortages hindered industrial development and further led to recession, which widened the wealth gap. Second, from a social perspective, conflicts over energy sources arose between different regions. Third, from an ecological perspective, people realised that the natural environment would be insufficient to provide all energy needs. Hence, people pay more attention to sustainable development that meets the needs of the present generation without compromising the ability of future generations to meet their own needs from economic, social, and environmental perspectives [86].

With urban regeneration from the 1960s, many cultural heritages were being adaptively reused to achieve sustainable development from economic and social dimensions. Because the reuse of cultural heritages can cultivate new functions for them, it can save spending on construction and develop suitable businesses/industries to strengthen the economy. It can also conserve the cultural context, which increases social acculturation and decreases conflicts. All of these benefits come from utilising cultural heritages to achieve the sustainable development of living space in terms of social and economic aspects. In addition, they show the significance of the continued sustainable existence of cultural heritages.

With the ecological environment increasingly impacted by industrialisation and modernisation, the 21st century is witnessing drastic climate changes that have led to an increased frequency of extreme weather events and natural disasters, which threaten the conservation of cultural heritages. For example, after 2019, under the influence of the outbreak of the COVID-19 pandemic, in order to prevent the spread of the epidemic, a large number of cultural heritage sites were forced to close, which greatly reduced the economic resources for conservation, and a large number of related conservation staff were laid off, which also greatly affected conservation. These ecological-related issues have shown that ecology is an important issue in cultural heritage conservation that must be emphasised nowadays and in the future, as shown in the "*Fuzhou declaration*" (2021) [87]. It is increasingly realised that by conserving the ecological wisdom carried by cultural heritages, the relationship between humans and nature can be coordinated to a certain extent. Cultural heritage conservation has to be closely aligned with sustainable goals of economic, social, and ecological development. Now, cultural heritage conservation considers not only the role of cultural heritages in the human-made environment (such as urban or rural space) but also considers their role in the entire universe involving human-made and natural environments related to all species. Additionally, with the conservation perspective expanded to the natural environment, preserving underwater cultural heritages also emerged as important, as shown in the "*Convention on the protection of the underwater cultural heritage*" (2001) [88].

### 2.4.3. Cultural Heritage Conservation Based on Multidisciplinary Collaborations

The 21st century has seen unprecedented multidisciplinary collaborations, better supporting the conservation of cultural heritages. From a multidisciplinary and interdisciplinary perspective, using various methods to conserve cultural heritages are increasingly emerging and emphasised. The methods are mainly employed in researching, protecting, managing, and reusing cultural heritages (see Table 1). Significantly, using computer technology and big data to conserve cultural heritage has also been highly valued and practised. This conservation is mainly in the 'smart city' context to control the relationship between cultural heritages and their surrounding environments for underpinning the conservation strategy to achieve the conservation of cultural heritages while also achieving sustainable urban development [89].

**Table 1.** The main methods for conserving cultural heritage.

| Method | Researching the Information of Cultural Heritages | | | | | | | | | Protecting Cultural Heritages | | | | Managing Cultural Heritages | | | | | | Reusing Cultural Heritages | |
|---|---|---|---|---|---|---|---|---|---|---|---|---|---|---|---|---|---|---|---|---|---|
| | Identifying age | Study of materials | Detect the degree of corrosion of the materials | Structural examination | Measure the form | Detect underground location and shape | Build models of the part of cultural heritage that cannot be seen | Explore the communication relationship between cultures | Evaluate their value | Establish a protective coating | Detect protecting coating or stains | Cleaning | Predict the impact of natural disasters | Record | Classification | Digital identify cultural heritages | Monitor their changes | Simulate management | Manage the relationship between cultural heritages and their surrounding environment | Exhibit | Improve tourism services |
| Spectroscopy [90–101] | ✔ | ✔ | ✔ | ✔ | | | | | | | ✔ | | | | | | | | | | |
| Radiocarbon dating [102] | ✔ | | | | | | | | | | | | | | | | | | | | |
| Photographic techniques [103–107] | | | | | ✔ | | | | | | | | | ✔ | | | ✔ | | | ✔ | |
| Ground-penetrating radar [108] | | | | | | ✔ | | | | | | | | | | | | | | | |
| NMR relaxation [109] | | | | | | ✔ | | | | | | | | | | | | | | | |
| Laser ablation [110] | | | | | | | | | | | | ✔ | | | | | | | | | |
| Nanocomposite [111] | | | | | | | | | | ✔ | | | | | | | | | | | |
| Semantic methods [112] | | | | | | | | ✔ | | | | | | | | | | | | | |
| Statistical method [113] | | | | | | | | | ✔ | | | | | | | | | | | | |
| Virtual reality [114–117] | | | | | | | | | | | | | | ✔ | | | | | | ✔ | |
| Augmented reality [118] | | | | | | | | | | | | | | ✔ | | | | | | ✔ | |
| Machine learning [119,120] | | | | | | | | | | | | | ✔ | | ✔ | | | | | | |
| Deep learning [121] | | | | | | | ✔ | | | | | | | | ✔ | ✔ | | | | | |
| Information retrieval [122] | | | | | | | | | | | | | | | ✔ | | | | | | |
| Random forest [123] | | | | | | | | | | | | | ✔ | | | | | | | | |
| Remote sensing [124] | | | | | | | | | | | | | | | | | | | ✔ | | |
| Geographic information system [125] | | | | | | | | | | | | | | | | | | | ✔ | | |
| Big data [89,126] | | | | | | | | | | | | | | | | | | | ✔ | | |
| Building information modelling [127] | | | | | | | | | | | | | | | | | | ✔ | | | |
| Social media [128] | | | | | | | | | | | | | | | | | | | | | ✔ |

Note: "✔" means the method (shown in the left of the table) are used for the purpose (shown in the top of the table).



With the development of such new methods for cultural heritage conservation, the potential drawbacks have begun to be emphasised. Considerable research has criticised the related conservation techniques, which may destroy the authenticity and integrity of cultural heritages when conserving them [129]. Hence, seeking non-destructive methods to conserve cultural heritages from the plethora of multidisciplinary techniques is an important current and future topic. Additionally, with the development of information technology, digital heritages have also recently been valued because they are a kind of carrier that can record human civilisation information [130]. With the development of digital data, it can be estimated that digital heritages will come to be more greatly valued in the future.

### 3. Trends in the Conservation of Cultural Heritages

A review of the development of the conservation of cultural heritages reveals a number of trends. Various trends in the conservation of cultural heritages have appeared in the course of its development. First, cultural heritage conservation has gradually evolved from restoration alone based on documents, as in the 18th century, to evaluation of the role cultural heritages might play in contemporary and future society, as in the 1960s. Evaluation not only provides a reason for the continued existence of cultural heritages, but also allows the possibility of their reuse to achieve sustainable urban regeneration, while preserving their authenticity and integrity.

Second, the scope of cultural heritage conservation has gradually expanded. In the beginning, cultural heritage conservation mainly focused on conserving heritages themselves. After the 1960s, conservation began to extend to their surroundings and even the urban context to coordinate conservation and urban development. In the 21st century, conserving cultural heritages has expanded from considering the human-made environment in terms of its economic-social importance to embracing the natural ecological environment and considering the harmony between heritages and nature.

Third, the range of cultural heritage conservation objects has gradually broadened. In the 18th century, conservation objects were mainly in the on archaeological field, including monuments, archaeological sites, and important buildings. In the 1960s, the role of cultural heritage in society became apparent, and various objects related to urban development began to be conserved, mainly historic cities, towns, villages, and industries. Intangible cultural heritage was also emphasised at that time. With the conservation of tangible and intangible cultural heritages, the spirit of places reflected by cultural heritages also began to be emphasised around 2000, and the spirit may affect people's consciousness. Nowadays, digital heritages and underwater cultural heritages are also being noticed.

Fourth, the vision of cultural heritage conservation is becoming more and more globalised with an increasing emphasis on diversity. In the 1950s, the international community mainly considered that cultural heritages should be preserved from being destroyed by wars. A decade after, conserving them through international cooperation was gradually highlighted. In the 1990s, it was noticed that conserving cultural heritages needed to consider their characteristics and local context. Now, conserving the diverse cultural heritages of the world in the name of justice and reducing social conflicts is highlighted.

Fifth, cultural heritage conservation increasingly respects human rights. Originally, cultural heritage conservation was mainly conducted by elites. In the 1960s, encouraging and educating public participation in cultural heritage conservation were strongly emphasised to enable the conservation to meet the needs of the public and conserve intangible cultural heritages carried by residents. Respecting human rights also reduces social conflicts brought by social injustice and maintains social stability.

Sixth, using multidisciplinary approaches to conserve cultural heritages is emphasised, and many non-destructive conserving methods are being proposed. In particular, using computer power to conserve cultural heritages has become a contemporary trend, allowing integrated conservation between cultural heritages and their surrounding context to be

conducted based on big data. Significantly, it needs to be noticed that technology needs to be used appropriately.

## 4. Conclusions

The contemporary principles of cultural heritage conservation can be roughly traced through these trends. They are as follows.

- Conserving cultural heritages firstly needs to conserve not only important monuments, historic sites, and historic buildings but also contexts associated with daily human life, including historic cities, towns, villages, and industries. Secondly, conservation needs to be concerned with cultural heritages not only on the land but also under water. Thirdly, conservation needs to focus on conserving not only tangible heritages but also intangible heritages, the spirit of places, and even digital heritages.
- Conserving cultural heritages includes not only preserving them by restoration but also establishing their value so that they can be reused and play a role in contemporary society. The restoration needs to be based on historical documents, and the historical information they embody must not be destroyed. Any elements added through restoration in different periods need to be distinguished from the original elements and be recorded and respected. Identifying the values of cultural heritages provides a basis for their sustainable conservation and allows them to contribute to society. Significantly, the relationship between conservation and reuse needs to be balanced in order not to impact their authenticity and integrity.
- The scope of cultural heritage conservation is not limited to conserving the heritages themselves but also involves conserving their surrounding environments to coordinate the relationship between them and their surroundings in terms of economic, social, and ecological aspects. Coordinating this relationship is necessary to conserve the heritages sustainably and promote the development of sustainable human-made and natural environments.
- When conserving cultural heritages, the power of the whole society needs to be mobilised if each group of stakeholders is to participate in the conservation, and if the needs of each group of stakeholders are to be met. Meanwhile, educating the public to conserve is important to enable people to participate in the conservation and increase the scientificity of the conservation.
- International cooperation to conserve cultural heritages is needed to conserve the diversity of human culture. Meanwhile, cultural heritages worldwide need to be conserved fairly, according to their local situation. This conserves cultural diversity and increases social stability by respecting every culture.
- Approaches to conserving cultural heritages using non-destructive methods can be considered from a multidisciplinary perspective. Significantly, their authenticity and integrity should not be destroyed by using inappropriate methods.

From the perspective of future conservation, three directions can be predicted. First, a series of phenomena have shown that the future will be a stage of frequent extreme climates and disasters [131]. Hence, how to conserve cultural heritages and prevent them from being impacted by disasters will gradually be highlighted. Second, with the gradual satisfaction of human's need for physical wealth, they will pay more and more attention to their spiritual world. Consequently, conserving the culture carried by cultural heritages and the role of culture in influencing people's consciousness will be gradually emphasised, leading to people's consciousness playing an active role in promoting the harmonious development of society. Third, using computer power for conserving cultural heritages will also become important. However, the only certainty in the future is uncertainty. Many possibilities in the future may affect conservation, which needs to be considered according to specific circumstances. Conserving authenticity and integrity of cultural heritages and allowing them to play a role in contemporary life will be an eternal subject.

These research findings would provide basic principles for cultural heritage conservation now and in the future. Historical experience should help to avoid problems such as

unclear conservation objects, scope, and methods, unbalanced relationships between conservation and utilisation, inadequate public participation in conservation, and insufficient conservation of the diversity of culture in the world.

**Author Contributions:** Conceptualisation, L.L. and Y.T.; methodology, L.L. and Y.T.; resources, L.L. and Y.T.; writing—original draft preparation, L.L.; writing—review and editing, L.L. and Y.T.; visualisation, L.L.; supervision, Y.T.; project administration, Y.T. All authors have read and agreed to the published version of the manuscript.

**Funding:** This research received no external funding.

**Conflicts of Interest:** The authors declare no conflicts of interest.

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
