# Peer review of "Towards the Contemporary Conservation of Cultural Heritages: An Overview of Their Conservation History"

_heritage, doi:10.3390/heritage7010009_

Round 1

Reviewer 1 Report

Comments and Suggestions for Authors

The research analyzes contemporary principles for the conservation of cultural heritage starting from its history since the 18th century. It investigates their practices, international theoretical approaches and related literature (the summary table is interesting). The aim is to highlight errors and good practices to lay the foundations for a conscious and correct conservation of cultural heritage throughout the contemporary era.

The importance of taking into account different classes of cultural assets that come from the daily life of man and which are both material and immaterial is underlined.

Furthermore, one of the aspects on which attention is focused concerns the context in which the cultural heritage is located with particular reference to urban contexts. Here we highlight how it is necessary to integrate cultural heritage conservation with the context of the sustainable development goals to successfully balance the relationship between the built and natural environment.

The reflection on the relationship with the public is interesting. An aspect that is present in the Faro Convention and which I suggest to include in the text, broadening the perspective of community involvement in the field of cultural heritage conservation.

This is in line with the idea that the conservation of cultural heritage is a global responsibility that involves joint cooperation by experts and communities that enjoy the cultural asset.

Finally, regarding the correct need to apply interdisciplinary methods of conservation of cultural heritage, maintaining that these must be limited to not destroying the authenticity and integrity of the cultural asset, I suggest also referring to Cesare Brandi's Restoration Theory and to the Athens Charter in the text.

Author Response

Dear Reviewer:

Thank you for giving me the opportunity to submit a revision of my manuscript, titled

"Towards the contemporary conservation of cultural heritages an overview of their conservation history" to
Journal of Heritage. We are grateful to the reviewers for their insightful comments on the manuscript. We have been able to incorporate changes to reflect suggestions provided by the reviewers. We have highlighted the changes below.

Comment 1: One of the aspects on which attention is focused concerns the context in which the cultural heritage is located with particular reference to urban contexts. Here we highlight how it is necessary to integrate cultural heritage conservation with the context of the sustainable development goals to successfully balance the relationship between the built and natural environment.

Response 1: Thank you for pointing this out. We have further explained the question as follows. Social and economic aspects: The reuse of cultural heritages can cultivate new functions for them, it can save spending on construction and develop suitable businesses/industries to strengthen the economy. It can also conserve the cultural context, which increases social acculturation and decreases conflicts. All of these benefits come from utilising cultural heritages to achieve the sustainable development of living space in terms of social and economic aspects. Ecological aspect: It is increasingly realised that by conserving the ecological wisdom carried by cultural heritages, the relationship between humans and nature can be coordinated to a certain extent.

Comment 2: The reflection on the relationship with the public is interesting. An aspect that is present in the Faro Convention and which I suggest to include in the text, broadening the perspective of community involvement in the field of cultural heritage conservation. This is in line with the idea that the conservation of cultural heritage is a global responsibility that involves joint cooperation by experts and communities that enjoy the cultural asset.

Response 2: We are grateful and agree for your suggestions. We explained that the cooperation by experts and communities in the conservation of cultural heritages can play the role of experts and communities in the conservation and enable the conservation to meet their needs.

Comment 3: Finally, regarding the correct need to apply interdisciplinary methods of conservation of cultural heritage, maintaining that these must be limited to not destroying the authenticity and integrity of the cultural asset, I suggest also referring to Cesare Brandi's Restoration Theory and to the Athens Charter in the text.

Response 3: We agree with this and have incorporated your suggestion throughout the

Manuscript. Based on Cesare Brandi's Restoration Theory we further explained that the restoration of cultural heritages needs to be based on historical evidence and to continue their artistry.

Additional clarifications

In addition to the above comments, the expression for improving the accuracy of the paper has been improved, and all spelling and grammatical errors pointed out by the reviewers have been corrected.

Sincerely,

Lanxin Li

12 December 2023

Reviewer 2 Report

Comments and Suggestions for Authors

The paper is correctly written and structured. It addresses a topic of great importance in today's world, since the concept of Cultural Heritage is being reviewed from different perspectives that allow us to approach its study, musealization, preventive conservation or restoration with criteria much more adapted to the present.

This publication contributes in two ways to this area of knowledge:

1- The exhaustive historographic review carried out, which dates back to the 17th century, and which includes, in addition to specialized bibliography, primary sources of great importance (such as "The Seven Lamps of Architecture" by J. Ruskin, Athens Charter, etc...)

2- Argue that in the 21st century the keys to guaranteeing the conservation of Heritage go far beyond what was considered until a few decades ago. The authors include at least four fundamental keys: (1) all types of objectives and manifestations must be considered; (2) we must consider the intangible dimension of heritage; (3) must consider interdisciplinary approaches; (4) VERY IMPORTANT: sustainability in the Conservation of Cultural Heritage depends on society; It is the local population itself that can promote the dynamics that guarantee the conservation and perpetuation of its heritage (which is equivalent to talking about cultural identity. Therefore, the awareness of local populations (of society) is a way to promote heritage conservation and sustainability.

For all the above, it is a contribution that has to be published. It provides evidence and proposals on the topic of great interest to the scientific community.

Author Response

Dear Editors and Reviewers:

Thank you for giving me the opportunity to submit a revision of my manuscript, titled

"Towards the contemporary conservation of cultural heritages an overview of their conservation history" to
Journal of Heritage. We are grateful to the reviewers for their insightful comments on the manuscript.

Comment 1: The exhaustive historographic review carried out, which dates back to the 17th century, and which includes, in addition to specialized bibliography, primary sources of great importance (such as "The Seven Lamps of Architecture" by J. Ruskin, Athens Charter, etc...)

Response 1: Thank you for your recognition of our manuscript and for giving out the opportunity to submit a revision of our manuscript.

Comment 2: Argue that in the 21st century the keys to guaranteeing the conservation of Heritage go far beyond what was considered until a few decades ago. The authors include at least four fundamental keys: (1) all types of objectives and manifestations must be considered; (2) we must consider the intangible dimension of heritage; (3) must consider interdisciplinary approaches; (4) VERY IMPORTANT: sustainability in the Conservation of Cultural Heritage depends on society; It is the local population itself that can promote the dynamics that guarantee the conservation and perpetuation of its heritage (which is equivalent to talking about cultural identity. Therefore, the awareness of local populations (of society) is a way to promote heritage conservation and sustainability.

Response 2: Thank you for your recognition of our manuscript and for giving out the opportunity to submit a revision of our manuscript.

Additional clarifications

In addition to the above comments, the expression for improving the accuracy of the paper has been improved, and all spelling and grammatical errors pointed out by the reviewers have been corrected.

Sincerely,

Lanxin Li

12 December 2023